# Evaluating Neuro-Symbolic AI Architectures: Design Principles, Qualitative Benchmark, Comparative Analysis and Results

**Oualid Bougzime**                                                     OUALID.BOUGZIME@UTBM.FR
*ICB UMR 6303 CNRS, Université Marie et Louis Pasteur, UTBM, F-90010 Belfort Cedex, France*

**Samir Jabbar**                                                       SAMIR.JABBAR@U-BOURGOGNE.FR
*ICB UMR 6303 CNRS, Université Bourgogne Europe, 21078 Dijon, France*

**Christophe Cruz**                                                       CHRISTOPHE.CRUZ@UBE.FR
*ICB UMR 6303 CNRS, Université Bourgogne Europe, F-21000 Dijon, France*

**Frédéric Demoly**                                                       FREDERIC.DEMOLY@UTBM.FR
*ICB UMR 6303 CNRS, Université Marie et Louis Pasteur, UTBM, F-90010 Belfort Cedex, France*

*Institut universitaire de France (IUF), Paris, France*

**Editors:** Leilani H. Gilpin, Eleonora Giunchiglia, Pascal Hitzler, and Emile van Krieken

## Abstract

Neuro-symbolic artificial intelligence (NSAI) represents a transformative approach in artificial intelligence (AI) by combining deep learning's ability to handle large-scale and unstructured data with the structured reasoning of symbolic methods. By leveraging their complementary strengths, NSAI enhances generalization, reasoning, and scalability while addressing key challenges such as transparency and data efficiency. This paper systematically studies diverse NSAI architectures, highlighting their unique approaches to integrating neural and symbolic components. This study then evaluates these architectures against comprehensive set of criteria, including generalization, reasoning capabilities, transferability, and interpretability, therefore providing a comparative analysis of their respective strengths and limitations. Notably, the Neuro $\rightarrow$ Symbolic $\leftarrow$ Neuro model consistenty outperforms its counterparts across all evaluation metrics. This result aligns with state-of-the-art research that highlight the efficacy of such architectures in harnessing advanced technologies like multi-agent systems. Moreover, our NSAI framework using retrieval-augmented illustrates how the 4D printing ontology can be systematically enriched with additional classes, object properties, data properties and individuals.

## 1. Introduction

Neuro-symbolic artificial intelligence (NSAI) is fundamentally defined as the combination of deep learning and symbolic reasoning (Garcez and Lamb, 2023). This hybrid approach aims to overcome the limitations of both symbolic and neural artificial intelligence (AI) systems while harnessing their respective strengths. Symbolic AI excels in reasoning and interpretability, whereas neural AI thrives in learning from vast amounts of data. By merging these paradigms, NSAI aspires to embody two fundamental aspects of intelligent cognitive behavior: the ability to learn from experience and the capacity to reason based on acquired knowledge (Valiant, 2003).

NSAI offers a promising avenue for addressing limitations of purely symbolic or neural systems. For instance, while neural networks (NNs) often struggle with interpretability,

symbolic AI systems are rigid and require extensive domain knowledge. By combining the adaptability of neural models with the explicit reasoning capabilities of symbolic methods, NSAI systems aim to provide enhanced generalization, interpretability, and robustness. These characteristics make NSAI particularly well-suited for solving complex, real-world problems where adaptability and transparency are critical (Hamilton et al., 2024). Kautz (Kautz, 2022) identifies several NSAI architectures that effectively integrate these paradigms, each architecture offers unique advantages but also poses specific challenges in terms of scalability, interpretability, and adaptability. A systematic evaluation of these architectures is imperative to understand their potential and limitations, guiding future research in this rapidly evolving field. The goal of evaluating NSAI architectures is to determine which ones are best suited to specific needs based on relevant criteria and to facilitate their adoption across various domains.

Therefore, this research aims to explore the core categories of NSAI and examine the insights that this classification yields regarding their strengths and limitations. The study is structured around three primary objectives: (i) to define, analyze and extend existing NSAI architectures, (ii) to develop a systematic framework for assessing these architectures across various criteria, and (iii) to demonstrate the *Symbolic → Neuro → Symbolic* paradigm in a practical 4D printing use case, illustrating how Large Language Models (LLMs) can be leveraged to enrich and expand a 4D printing ontology.

## 2. Neuro-Symbolic AI Architectures

This section provides an overview of various NSAI architectures, offering insights into their design principles, integration strategies, and unique capabilities. While Kautz's classification serves as a foundational framework, we extend it by incorporating additional architectural perspectives—most notably the fibring architecture—and by introducing more granular subclassifications within the compiled architecture to capture the evolving landscape of NSAI systems. This expanded categorization highlights the diversity of design strategies and the broad applicability of NSAI techniques, emphasizing their potential for more interpretable, robust, and data-efficient AI solutions.

### 2.1. Sequential

As part of the sequential NSAI, the *Symbolic → Neuro → Symbolic* architecture involves systems where both the input and output are symbolic, with a NN acting as a mediator for processing. Symbolic input, such as logical expressions or structured data, is first mapped into a continuous vector space through an encoding process. The NN operates on this encoded representation, enabling it to learn complex transformations or patterns that are difficult to model symbolically. Once the processing is complete, the resulting vector is decoded back into symbolic form, ensuring that the final output aligns with the structure and semantics of the input domain. A formulation of this architecture is presented below:

$$y = f_{\text{neural}}(x) \tag{1}$$

where $x$ is the symbolic input, $f_{\text{neural}}(x)$ represents the NN that processes the input, and $y$ is the symbolic output. This architecture can be used in a semantic parsing task, where

the input is a sequence of symbolic tokens (e.g., words). Here, each token is mapped to a continuous embedding via word2vec or a similar method (Mikolov, 2013; Pennington et al., 2014). The NN then processes these embeddings to learn compositional patterns or transformations. From this, the network's output layer decodes the processed information back into a structured logical form (eg. knowledge-graph triples).

## 2.2. Nested

The nested NSAI category is composed of two different architectures. The first – *Symbolic[Neuro]* – places a NN as a subcomponent within a predominantly symbolic system. Here, the NN is used to perform tasks that require statistical pattern recognition, such as extracting features from raw data or making probabilistic inferences, which are then utilized by the symbolic system. This architecture can formally defined as follows:

$$y = g_{\text{symbolic}}(x, f_{\text{neural}}(z)) \tag{2}$$

where $x$ represents the symbolic context, $z$ is the input passed from the symbolic reasoner to the NN, $f_{\text{neural}}(z)$ expresses the neural model processing the input, and $g_{\text{symbolic}}$ the symbolic reasoning engine that integrates neural outputs. A well-known instance of this architecture is AlphaGo (Silver et al., 2016), where a symbolic Monte-Carlo tree search orchestrates high-level decision-making, while a NN evaluates board states, providing a data-driven heuristic to guide the symbolic search process (Coulom, 2006).

The second architecture – *Neuro[Symbolic]* – integrates a symbolic reasoning engine as a component within a neural system, allowing the network to incorporate explicit symbolic rules or relationships during its operation. The symbolic engine provides structured reasoning capabilities, such as rule-based inference or logic, which complement the NN's ability to generalize from data. By embedding symbolic reasoning within the neural framework, the system gains interpretability and structured decision-making while retaining the flexibility and scalability of neural computation. This integration is particularly effective for tasks that require reasoning under constraints or adherence to predefined logical frameworks (Heule et al., 2016). This configuration can be described as follows:

$$y = f_{\text{neural}}(x, g_{\text{symbolic}}(z)) \tag{3}$$

where $x$ represents the input data to the neural system, $z$ is the input passed from the NN to the symbolic reasoner, $g_{\text{symbolic}}$ is the symbolic reasoning function, and $f_{\text{neural}}$ denotes the NN processing the combined inputs. This framework can be exemplified by considering a scenario where a symbolic reasoning engine processes structured data—such as a maze—to generate a solution path. In this example, a neural network encodes the problem into a latent representation and subsequently decodes it into a symbolic sequence of actions (e.g., forward, turn left, turn right).

## 2.3. Cooperative

As a cooperative framework, *Neuro | Symbolic* uses neural and symbolic components as interconnected coroutines, collaborating iteratively to solve a task. NNs process unstructured data, such as images or text, and convert it into symbolic representations that are

easier to reason about. The symbolic reasoning component then evaluates and refines these representations, providing structured feedback to guide the NN's updates. This feedback loop continues over multiple iterations until the system converges on a solution that meets predefined symbolic constraints or criteria. By combining the strengths of NNs for generalization and symbolic reasoning for interpretability, this approach achieves robust and adaptive problem-solving (Mao et al., 2019). This architecture can be described as follows:

$$z^{(t+1)} = f_{\text{neural}}(x, y^{(t)}), \quad y^{(t+1)} = g_{\text{symbolic}}(z^{(t+1)}), \quad \forall t \in \{0, 1, \ldots, n\} \tag{4}$$

where $x$ represents non-symbolic data input, $z^{(t)}$ is the intermediate symbolic representation at iteration $t$, $y^{(t)}$ is the symbolic reasoning output at iteration $t$, $f_{\text{neural}}(x, y^{(t)})$ expresses the NN that processes the input $x$ and feedback from the symbolic output $y^{(t)}$, $g_{\text{symbolic}}(z^{(t+1)})$ is the symbolic reasoning engine that updates $y^{(t+1)}$ based on the neural output $z^{(t+1)}$, and $n$ is the maximum number of iterations or a convergence threshold. The hybrid reasoning halts when the outputs $y^{(t)}$ converge (e.g., $|y^{(t+1)} - y^{(t)}| < \epsilon$)), where $\epsilon$ is a small threshold denoting minimal change between successive outputs, or when the maximum iterations $n$ is reached. For instance, this architecture can applied in autonomous driving systems, where a NN processes real-time images from vehicle cameras to detect and classify traffic signs. It identifies shapes, colors, and patterns to suggest potential signs, such as speed limits or stop signs. A symbolic reasoning engine then evaluates these detections based on contextual rules—like verifying if a detected speed limit sign matches the road type or if a stop sign appears in a logical position (e.g., near intersections). If inconsistencies are detected, such as a stop sign identified in the middle of a highway, the symbolic system flags the issue and prompts the neural network to re-evaluate the scene. This iterative feedback loop continues until the system reaches consistent, high-confidence decisions, ensuring robust and reliable traffic sign recognition, even in challenging conditions like poor lighting or partial occlusions.

### 2.4. Compiled

As part of the compiled NSAI, $Neuro_{Symbolic_{Loss}}$ uses symbolic reasoning into the loss function of a NN. The loss function is typically used to measure the discrepancy between the model's predictions and the true outputs. By incorporating symbolic rules or constraints, the network's training process not only minimizes prediction error but also ensures that the output aligns with symbolic logic or predefined relational structures. This allows the model to learn not just from data but also from symbolic reasoning, helping to guide its learning process toward solutions that are both accurate and consistent with symbolic principles.

$$\mathcal{L} = \mathcal{L}_{\text{task}}(y, y_{\text{target}}) + \lambda \cdot \mathcal{L}_{\text{symbolic}}(y) \tag{5}$$

where $y$ is the model prediction, $y_{\text{target}}$ represents the ground truth labels, $\mathcal{L}_{\text{task}}$ is the task-specific loss (e.g., cross-entropy), $\mathcal{L}_{\text{symbolic}}$ is the penalization for violating symbolic rules, $\lambda$ the Weight balancing the two loss components, and $\mathcal{L}$ the final loss, combining both the task-specific loss and the symbolic constraint penalty to guide model optimization. This architecture is typically useful in the field of 4D printing, where structures need to be optimized at the material level to achieve a target shape. In such a case, a NN predicts the material distribution and geometric configuration that allows the structure to adapt under external stimuli. The training process incorporates a physics-informed loss function, where,

in addition to minimizing the difference between predicted and desired mechanical behavior, the model is penalized whenever the predicted deformation violates symbolic mechanical constraints, such as equilibrium equations or the stress-strain relationship.

A second compiled NSAI architecture, called $Neuro_{Symbolic_{Neuro}}$, uses symbolic reasoning at the neuron level by replacing traditional activation functions with mechanisms that incorporate symbolic reasoning. Rather than using standard mathematical operations like ReLU or sigmoid, the neuron activation is governed by symbolic rules or logic. This allows the NN to reason symbolically at a more granular level, integrating explicit reasoning steps into the learning process. This architecture can be described as follows:

$$y = g_{\text{symbolic}}(x) \tag{6}$$

where: $x$ represents the pre-activation input, $g_{\text{symbolic}}(x)$ is the symbolic reasoning-based activation function, and $y$ the final neuron. This architecture can find application in lean approval systems, where neural activations are driven by symbolic financial rules rather than traditional functions. One example is the collateral-based constraint neuron, which dynamically adjusts the risk score based on the value of the pledged collateral. When the collateral's value falls below a predefined threshold relative to the loan amount, the neuron applies a strict penalty that substantially increases the risk score, effectively preventing the system from underestimating the associated financial risk.

Finally, the last compiled architecture, *Neuro:Symbolic → Neuro*, uses a symbolic reasoner to generate labeled data pairs $(x, y)$, where $y$ is produced by applying symbolic rules or reasoning to the input $x$. These pairs are then used to train a NN, which learns to map from the symbolic input $x$ to the corresponding output $y$. The symbolic reasoner acts as a supervisor, providing high-quality, structured labels that guide the NN's learning process (Riegel et al., 2020). This architecture can be governed as follows:

$$\mathcal{D}_{\text{train}} = \{(x, g_{\text{symbolic}}(x)) \mid x \in \mathcal{X}\} \tag{7}$$

where $\mathcal{D}_{\text{train}}$ is the training dataset, $x$ denotes the unlabeled data, $g_{\text{symbolic}}(x)$ represents symbolic rules generating labeled data, and $\mathcal{X}$ the set of all input data.

## 2.5. Ensemble

Another promising architecture, called *Neuro → Symbolic ← Neuro* uses multiple interconnected NNs via a symbolic fibring function, which enables them to collaborate and share information while adhering to symbolic constraints. The symbolic function acts as an intermediary, facilitating communication between the networks by ensuring that their interactions respect predefined symbolic rules or structures. This enables the networks to exchange information in a structured manner, allowing them to jointly solve problems while benefiting from both the statistical learning power of NNs and the logical constraints imposed by the symbolic system (Garcez and Gabbay, 2004). This architecture can formally defined as follows:

$$y = g_{\text{fibring}}(\{f_i\}_{i=1}^n) \tag{8}$$

where $f_i$ represents the individual NN, $g_{\text{fibring}}$ is the logic-aware aggregator that enforces symbolic constraints while unifying the outputs of multiple NNs, $n$ the number of NNs, and

$y$ is the combined output of interconnected NNs, produced through the symbolic fibring function $g_{\text{fibring}}$. For instance, two NNs communicate through activation states, which enables dynamic exchange of learned representations (Garcez and Gabbay, 2004).

Finally, we substantially advance Hertz's foundational taxonomy by introducing two pioneering paradigms: the Neuro$_{\text{Symbolic}_{\text{Loss}}}$ architecture, which embeds symbolic constraints directly into the neural loss function, and a fibring-based ensemble mechanism that orchestrates multiple neural modules via a logic-aware aggregator to enforce explicit symbolic rules. Beyond these extensions, we deliver an unprecedentedly granular taxonomy for each architectural family, detailing their integration pathways, iterative dynamics, and unique trade-offs.

## 3. Evaluation of NSAI Architectures

Ensuring the reliability and practical applicability of NSAI architectures requires a systematic evaluation across multiple well-defined criteria. Such an evaluation not only identifies the strengths and limitations of the architectures but also fosters trust among stakeholders by emphasizing interpretability, transparency, and robustness—qualities essential in domains such as healthcare, finance, and autonomous systems. Moreover, a rigorous assessment provides benchmarks that can stimulate the development of next-generation models. The following sections delineate the key criteria for evaluating NSAI architectures.

### 3.1. Core Criteria

NSAI architectures are evaluated on several key aspects and each of these high-level categories is further subdivided into subcategories that specify the precise dimensions and metrics employed for rigorous assessment. **Generalization**—the capacity to extend learned representations beyond the training dataset to perform effectively in novel or unforeseen situations, which is assessed by *out-of-distribution performance* (Yang et al., 2024), meaning the model maintains accuracy when inputs deviate from the training distribution; by *contextual flexibility* (Patel et al., 2015), where it adapts seamlessly to new domains or tasks with minimal retraining; and by *relational accuracy* (Ye et al., 2024), capturing its ability to identify genuine dependencies in data while suppressing spurious correlations; **Scalability**—the ability to sustain efficiency as data volumes and computational demands increase, which includes *large-scale adaptation* (Dean et al., 2012), where the system processes and derives insights from massive datasets end-to-end; *hardware efficiency* (Silvano et al., 2023), which optimizes computational resource utilization across both low-resource devices and high-performance infrastructures; and *complexity management* (Tan and Le, 2019), meaning the architecture can grow in depth or breadth without incurring prohibitive latency or deployment overhead; **Data Efficiency**—the capacity to learn effectively from limited data, assessed by *data reduction* (Song et al., 2023), which achieves target performance with fewer labeled examples; by *data optimization* (Zhu, 2005), where unlabeled or weakly labeled data are leveraged through semi- or self-supervised learning; and by *incremental adaptability* (Gunasekara et al., 2023), meaning new samples can be incorporated on the fly without requiring a full retraining cycle; **Reasoning**—the integration of neural pattern recognition and symbolic manipulation to derive logical conclusions, evaluated by *logical reasoning* (Hou, 2025), in which explicit rules are systematically applied for precise

inference; by *relational understanding* (Li et al., 2025), where complex inter-entity relationships are comprehended; and by *cognitive versatility* (Ke et al., 2025), meaning the model combines deductive, inductive and abductive paradigms to address a wide range of problem classes; **Robustness**—the resilience to noise, adversarial perturbations and dynamic environments, assessed by *perturbation resistance* (Meng et al., 2022), where stable outputs are maintained under noisy or malicious inputs; by *adaptive resilience* (Liu et al., 2024), meaning functionality endures amid shifting or unpredictable conditions; and by *bias resilience* (Hort et al., 2024), in which systematic errors are detected and mitigated to uphold fairness and accuracy; **Transferability**—the ability to apply acquired knowledge across different domains and tasks with minimal overhead, evaluated by *multi-domain adaptation* (Zhou et al., 2022), where representations generalize across disparate data regimes; by *multi-task learning* (Zhang and Yang, 2021), in which heterogeneous objectives are handled concurrently via shared representations; and by *personalization* (Zhang et al., 2024), meaning fine-grained adjustments meet individual user or application requirements; and **Interpretability**—ensuring transparent and trustworthy decision processes, assessed by *transparency* (Lipton, 2018), which reveals the clarity of internal mechanisms; by *explanation* (Ribeiro et al., 2016), where the model generates comprehensible justifications for its outputs; and by *traceability* (Spoczynski et al., 2025), meaning the full sequence of operations and feature contributions underlying each decision can be reconstructed.

## 3.2. Evaluation Methodology

The evaluation of NSAI architectures was conducted using a systematic approach to ensure a robust and transparent assessment of their performance across multiple criteria. This process relied on three key sources: scientific literature, empirical findings, and an analysis of the design principles underlying each architecture. The scientific literature served as the primary source of qualitative insights, offering detailed analyses of the strengths and limitations of various architectures; the specific studies consulted are listed in Appendix A.

Foundational research and state-of-the-art studies provided evidence of performance in areas such as scalability, reasoning, and interpretability, helping to guide the evaluation. Additionally, empirical results from experimental studies and benchmarks offered quantitative data, enabling objective comparisons across architectures. Metrics such as accuracy, adaptability, and efficiency were particularly valuable in validating the claims made in research papers. The design principles of each technology were also considered to understand how neural and symbolic components were integrated. This analysis provided insights into the inherent capabilities and constraints of each architecture, such as its suitability for handling complex reasoning tasks, scalability to large datasets, or adaptability to dynamic environments.

For each main criterion, architectures are rated on a four-point scale according to how many of its three sub-criteria they satisfy: if all three sub-criteria are met, the rating is *High*, reflecting consistently performance; if two are met, the rating is *Medium*, indicating generally satisfactory results with some limitations; if only one sub-criterion is met, the rating is *Low–Medium*, denoting limited strengths; and if none are met, the rating is *Low*, signifying significant weaknesses or inconsistent outcomes. In this way, each architecture receives a quantitative score ranging from 0 to 3, ensuring a balanced and evidence-based

evaluation. It provides a clear understanding of the strengths and weaknesses of each architecture, enabling meaningful comparisons and guiding future advancements in NSAI research and applications.

### 3.3. Results and Discussion

Appendix B provides a comparative analysis of various NSAI architectures across seven main evaluation criteria and their respective sub-criteria. This comprehensive evaluation highlights the strengths and weaknesses of each architecture.

Overall, the *Neuro → Symbolic ← Neuro* architecture emerges as the best-performing model, consistently achieving high ratings across all criteria. Its exceptional performance in generalization, scalability, and interpretability makes it highly suitable for real-world applications that demand reliability, adaptability, and transparency. While other architectures also perform well in specific areas, the versatility and robustness of *Neuro → Symbolic ← Neuro* set it apart as the most balanced and capable solution. This conclusion aligns with findings in the state of the art, which highlight the effectiveness of *Neuro → Symbolic ← Neuro* architectures in leveraging advanced AI technologies, such as multi-agent systems (MAS). MAS are well-documented for their robustness, particularly in dynamic and distributed environments, where their ability to coordinate, adapt, and reason collectively enables superior performance. For instance, (Subramanian et al., 2024) demonstrated that incorporating neuro-symbolic approaches into multi-agent RL enhances both interpretability and probabilistic decision-making. This makes such systems highly robust in environments with partial observability or uncertainties. These attributes are particularly valuable in *Neuro → Symbolic ← Neuro* architectures, as they address the critical need for transparency and robustness in complex real-world applications.

### 3.4. Ontology Application

The rapid advancements in 4D printing have introduced a need for structured frameworks to manage and formalize the diverse knowledge involved in designing transformable systems, and the HERMES ontology (Dimassi et al., 2021) addresses this by providing a semantic and logical foundation—built upon the Basic Formal Ontology (BFO)—for representing spatiotemporal transformations, material behaviors, and additive-manufacturing processes. While ontologies like HERMES offer a rigorous, interoperable schema for encoding domain knowledge, they remain largely static and ill-suited to unstructured text or evolving datasets. In contrast, LLMs excel at processing and generating human-like text and can dynamically complete ontological entries (e.g., suggesting "mechanical property" in a triple such as (Material, has, ?)) based on contextual cues. Recent multimodal LLMs (e.g., GPT-4V (Achiam et al., 2023)) further extend this capability by jointly reasoning over text and images, thereby enabling ontologies to adapt in real time to new discoveries and heterogeneous inputs.

Using the *Symbolic → Neuro → Symbolic* architecture, our RA-MLLMs framework bridges unstructured sources and structured knowledge by tightly coupling a fine-tuned multimodal LLM with ontology-driven reasoning. As depicted in Figure 1, published articles are first split into individual textual sections and extracted figures, each encoded into dense vectors and indexed in a high-performance retrieval store, while material datasets undergo

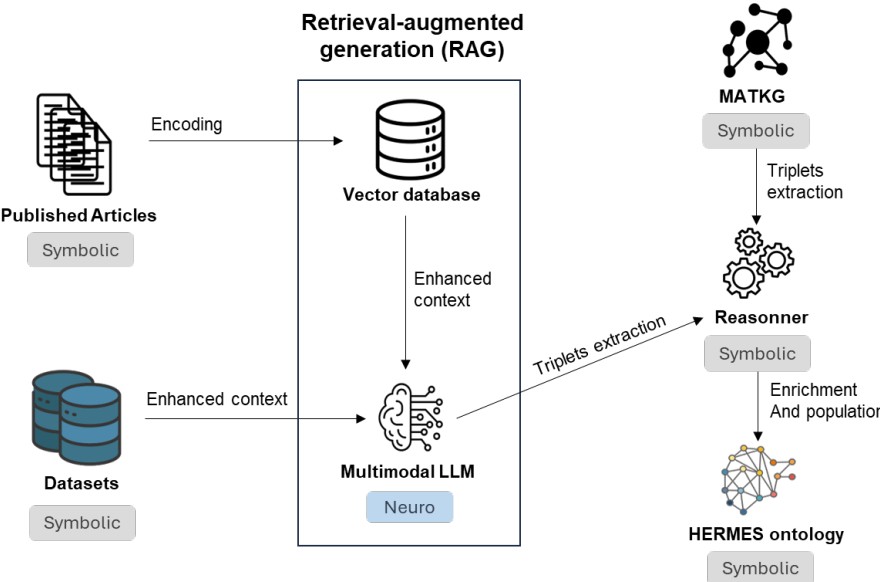

Figure 1: Retrieval-augmented MLLMs (RA-MLLMs) architecture

a column-centric pipeline—column names and descriptions are parsed and fed into the LLM via a one-shot prompting technique to map each field to an ontology class (instantiating each row as an instance of its corresponding class) and to identify relationships among these fields. At inference, the MLLM retrieves the most relevant text or image snippets and produces context-aware outputs, from which a dedicated triplet-extraction module pulls candidate triples. These newly extracted triples, together with existing entries from the MATKG knowledge graph (Venugopal and Olivetti, 2024), are submitted to a downstream symbolic reasoner, which performs rigorous validation (coherence, semantic consistency, structural checks and duplicate elimination) and ontology construction before populating and enriching the HERMES ontology. This continuous $Symbolic \rightarrow Neuro \rightarrow Symbolic$ loop ensures real-time knowledge updates and, through the LLM's generative and multimodal power, delivers enriched insights for the design and optimization of intelligent material systems.

Table 1 presents a side-by-side comparison of the original HERMES ontology and the three stages of our automated enrichment pipeline—article mining (based on a corpus of 1,500 articles), dataset (from 8 materials databases (Jain et al., 2013a; Kuenneth and Ramprasad, 2022; hyd, 2023; Crews et al., 2012; of Chicago, 2023; Jain et al., 2013b; Takahashi et al., 2024; NASA)) extraction, and MATKG integration—as well as the aggregated total of enriched entities contributed by each source, which exceeds 12.5 million, most of which originate from dataset extraction due to the high density of structured, field-level information in curated material databases. The quality of the extracted triplets is underpinned by a Graph BERTScore F1 (Saha et al., 2021) of 0.7, demonstrating high semantic fidelity. This enriched ontology then dramatically enhances knowledge discovery and underpins a fully automated design-to-manufacturing workflow, thereby accelerating innovation in 4D printing

([Bougzime et al.](), [2025]). Furthermore, it provides a robust foundation for explainable-AI decision-support systems, enabling domain experts to justify and transparently interpret every inference drawn from the ontology.

Table 1: HERMES ontology vs. RA-MLLMs framework results

| | HERMES Ontology | Extended Ontology | | | Summary |
| --- | --- | --- | --- | --- | --- |
| | | Article | Dataset | MATKG | |
| *Classes* | 170 | 5,706 | 144 | - | 5,849 |
| *Object properties* | 48 | 1,331 | 26 | - | 1,357 |
| *Data properties* | 13 | 4,390 | - | 2 (+ 445,370 relations) | 4,392 |
| *Instances* | 9 | 16,651 | 12,540,671 | 6,629 | 12,563,951 |

## 4. Conclusion

This study evaluates several NSAI architectures against a comprehensive set of criteria. Among the architectures investigated, Fibring architecture emerges as the most balanced and robust solution. It consistently demonstrates superior performance across multiple criteria. These results align with recent advancements in the field, which emphasize the role of multi-agent systems in enhancing robustness and adaptability. Future work will be focused on exploring the scalability of this architecture in even larger and more diverse environments. Additionally, advancing the integration of symbolic reasoning within multi-agent systems may further enhance their robustness and cognitive versatility. As the field evolves, $Neuro \rightarrow Symbolic \leftarrow Neuro$ architectures are likely to remain at the forefront of innovation, offering practical and scientifically grounded solutions to the most pressing challenges in AI.

## Acknowledgments

This research was funded by the IUF, Innovation Chair on 4D Printing, the French National Research Agency under the "France 2030 Initiative" and the "DIADEM Program", grant number 22-PEXD-0016 ("ARTEMIS").

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

## Appendix A.

Table 2 summarizes the key scientific articles we used to evaluate each architecture.

## Appendix B.

Table 3 presents the benchmark results for the Sequential, Ensemble and Cooperative architectures; Table 4 details the performance of the Nested architecture; and Table 5 summarizes the outcomes for the Compiled architecture.

Table 2: Set of relevant published NSAI architectures considered in the proposed study

| Architecture | References |
| --- | --- |
| $Symbolic \rightarrow Neuro \rightarrow Symbolic$ | (Kouris et al., 2021), (Sutherland et al., 2019), (Gu et al., 2019), (Cui et al., 2021), (Xu and Li, 2019), (Cowen-Rivers et al., 2019), (Bounabi et al., 2021), (Es-Sabery et al., 2021), (Lima et al., 2019), (Zhou et al., 2021), (Gong et al., 2020), (Tato et al., 2019), (Langton and Srihasam, 2021), (Braşoveanu and Andonie, 2019), (Pinhanez et al., 2021), (Dehua et al., 2021), (Fazlic et al., 2019), (D'Souza et al., 2019), (Ayyanar et al., 2019), (Hu et al., 2021), (Chen et al., 2020b), (Manda et al., 2020), (Honda and Hagiwara, 2019), (Schon et al., 2019), (Amin, 2019) |
| $Neuro[Symbolic]$ | (Heule et al., 2016), (Madan et al., 2021) |
| $Symbolic[Neuro]$ | (Silver et al., 2016), (Chen et al., 2021a), (Chen et al., 2021b), (Pacheco and Goldwasser, 2021), (Chaturvedi et al., 2019), (Qin et al., 2021) |
| $Neuro \mid Symbolic$ | (Mao et al., 2019), (Yao et al., 2018), (Shi et al., 2021), (Škrlj et al., 2021), (Wang and Pan, 2021), (Lemos et al., 2020), (Huang et al., 2019) |
| $Neuro \rightarrow Symbolic \leftarrow Neuro$ | (Das et al., 2021), (Garcez and Gabbay, 2004), (Belle et al., 2023), (Guo et al., 2025), (Jiang et al., 2024), (Guo et al., 2024), (Maldonado et al., 2024), (He, 2024), (Lo et al., 2024) |
| $Neuro:Symbolic \rightarrow Neuro$ | (Lample and Charton, 2019), (Yabloko, 2020), (Zhou et al., 2020), (Saveleva et al., 2021), (Gupta et al., 2021), (Demeter and Downey, 2020), (Jiang et al., 2021), (Kogkalidis et al., 2020), (Zhang et al., 2021), (Sen et al., 2020), (Huo et al., 2019), (Jiang et al., 2020), (Liu et al., 2021), (Chaudhury et al., 2021), (Verga et al., 2020), (Socher et al., 2013) |
| $Neuro_{Symbolic_{Loss}}$ | (Serafini and Garcez, 2016), (Raissi et al., 2019), (Chen et al., 2020a), (Graziani et al., 2019), (Altszyler Lemcovich et al., 2020), (Hussain and Cambria, 2018) |
| $Neuro_{Symbolic_{Neuro}}$ | (Smolensky et al., 2016) (Smolensky, 1990), (Xu et al., 2023), (Marra et al., 2019), (Shi et al., 2020), (Dragone et al., 2021), (Riegel et al., 2020), (West et al., 2021) |

Table 3: Comparison of NSAI architectures (Part 1): Sequentiel, Ensemble and Cooperative

| Main Criterion | Sub-Criterion | Symbolic Neuro Symbolic | Neuro → Symbolic ← Neuro | Neuro\|Symbolic |
|---|---|---|---|---|
| **Generalization** | Out-of-dist. | yes (Honda and Hagiwara, 2019) | yes (Belle et al., 2023) | yes (Yao et al., 2018) |
| | Continuous flex. | yes (Kouris et al., 2021) | yes (Guo et al., 2025) | yes (Lemos et al., 2020) |
| | Relative prec. | yes (Fazlic et al., 2019) | yes (Garcez and Gabbay, 2004) | no |
| | Summary | High | High | Medium |
| **Scalability** | Large-scale adapt. | yes (Cowen-Rivers et al., 2019) | yes (Lo et al., 2024) | yes (Škrlj et al., 2021) |
| | Hardware efficiency | no | yes (Jiang et al., 2024) | no |
| | Complexity | yes (Brașoveanu and Andonie, 2019) | yes (Maldonado et al., 2024) | no |
| | Summary | Medium | High | Low–Medium |
| **Data Efficiency** | Reduction | no | yes (He, 2024) | yes (Mao et al., 2019) |
| | Optimization | yes (Manda et al., 2020) | yes (Das et al., 2021) | yes (Škrlj et al., 2021) |
| | Incremental adapt. | yes (Pinhanez et al., 2021) | yes (Guo et al., 2024) | no |
| | Summary | Medium | High | Medium |
| **Reasoning** | Logical reason. | yes (Ayyanar et al., 2019) | yes (Guo et al., 2025) | yes (Shi et al., 2021) |
| | Comprehension | yes (Hu et al., 2021) | yes (Maldonado et al., 2024) | yes (Huang et al., 2019) |
| | Versatility | yes (Zhou et al., 2021) | yes (Garcez and Gabbay, 2004) | yes (Wang and Pan, 2021) |
| | Summary | High | High | High |
| **Robustness** | Perturbations | no | yes (Guo et al., 2025) | yes (Škrlj et al., 2021) |
| | Adaptability | yes (Pinhanez et al., 2021) | yes (He, 2024) | yes (Mao et al., 2019) |
| | Bias handling | no | yes (Guo et al., 2024) | no |
| | Summary | Low–Medium | High | Medium |
| **Transferability** | Multi-domain | yes (Sutherland et al., 2019) | yes (Maldonado et al., 2024) | yes (Lemos et al., 2020) |
| | Multi-task | yes (Dehua et al., 2021) | yes (Jiang et al., 2024) | no |
| | Personalization | yes (Es-Sabery et al., 2021) | yes (Guo et al., 2024) | no |
| | Summary | High | High | Low–Medium |
| **Interpretability** | Transparency | yes (Lima et al., 2019) | yes (Belle et al., 2023) | yes (Lemos et al., 2020) |
| | Explanation | yes (Langton and Srihasam, 2021) | yes (Garcez and Gabbay, 2004) | yes (Wang and Pan, 2021) |
| | Traceability | yes (Amin, 2019) | yes (Das et al., 2021) | yes (Shi et al., 2021) |
| | Summary | High | High | High |

Table 4: Comparison of NSAI architectures (Part 2): Nested

| Main Criterion | Sub-Criterion | Symbolic[Neuro] | Neuro[Symbolic] |
|---|---|---|---|
| **Generalization** | Out-of-dist. | no | yes (Madan et al., 2021) |
| | Continuous flex. | yes (Chen et al., 2021a) | no |
| | Relative prec. | yes (Qin et al., 2021) | no |
| | Summary | Medium | Low–Medium |
| **Scalability** | Large-scale adapt. | no | yes (Heule et al., 2016) |
| | Hardware efficiency | no | no |
| | Complexity | no | no |
| | Summary | Low | Low–Medium |
| **Data Efficiency** | Reduction | yes (Chen et al., 2021b) | yes (Madan et al., 2021) |
| | Optimization | yes (Pacheco and Goldwasser, 2021) | yes (Madan et al., 2021) |
| | Incremental adapt. | yes (Silver et al., 2016) | no |
| | Summary | High | Medium |
| **Reasoning** | Logical reason. | yes (Chen et al., 2021a) | yes (Heule et al., 2016) |
| | Comprehension | yes (Chaturvedi et al., 2019) | no |
| | Versatility | yes (Qin et al., 2021) | no |
| | Summary | High | Low–Medium |
| **Robustness** | Perturbations | yes (Silver et al., 2016) | yes (Madan et al., 2021) |
| | Adaptability | yes (Chaturvedi et al., 2019) | no |
| | Bias handling | no | no |
| | Summary | Medium | Low–Medium |
| **Transferability** | Multi-domain | yes (Chaturvedi et al., 2019) | no |
| | Multi-task | no | no |
| | Personalization | no | no |
| | Summary | Low–Medium | Low |
| **Interpretability** | Transparency | yes (Chen et al., 2021a) | yes (Madan et al., 2021) |
| | Explanation | yes (Chen et al., 2021b) | yes (Heule et al., 2016) |
| | Traceability | yes (Qin et al., 2021) | yes (Madan et al., 2021) |
| | Summary | High | High |

Table 5: Comparison of NSAI architectures (Part 3): Compiled

| Main Criterion | Sub-Criterion | Neuro:Symbolic → Neuro | Neuro$_{Symbolic_{Loss}}$ | Neuro$_{Symbolic_{Neuro}}$ |
|---|---|---|---|---|
| **Generalization** | Out-of-dist. | yes (Socher et al., 2013) | no | no |
| | Continuous flex. | no | no | no |
| | Relative prec. | no | no | no |
| | Summary | Low–Medium | Low | Low |
| **Scalability** | Large-scale adapt. | yes (Huo et al., 2019) | yes (Hussain and Cambria, 2018) | yes (Smolensky, 1990) |
| | Hardware efficiency | yes (Lample and Charton, 2019) | yes (Serafini and Garcez, 2016) | yes (Xu et al., 2023) |
| | Complexity | no | no | no |
| | Summary | Medium | Medium | Medium |
| **Data Efficiency** | Reduction | no | yes (Raissi et al., 2019) | yes (Smolensky et al., 2016) |
| | Optimization | yes (Gupta et al., 2021) | yes (Graziani et al., 2019) | yes (Marra et al., 2019) |
| | Incremental adapt. | no | no | no |
| | Summary | Low–Medium | Medium | Medium |
| **Reasoning** | Logical reason. | yes (Kogkalidis et al., 2020) | yes (Serafini and Garcez, 2016) | yes (Smolensky et al., 2016) |
| | Comprehension | yes (Liu et al., 2021) | yes (Chen et al., 2020a) | yes (Smolensky, 1990) |
| | Versatility | yes (Jiang et al., 2021) | no | no |
| | Summary | High | Medium | Medium |
| **Robustness** | Perturbations | yes (Zhou et al., 2020) | no | yes (Shi et al., 2020) |
| | Adaptability | yes (Sen et al., 2020) | no | no |
| | Bias handling | yes (Verga et al., 2020) | no | yes (Dragone et al., 2021) |
| | Summary | High | Low | Medium |
| **Transferability** | Multi-domain | no | no | no |
| | Multi-task | no | yes (Hussain and Cambria, 2018) | yes (Smolensky et al., 2016) |
| | Personalization | no | no | no |
| | Summary | Low | Low–Medium | Low–Medium |
| **Interpretability** | Transparency | yes (Saveleva et al., 2021) | yes (Altszyler Lemcovich et al., 2020) | yes (Smolensky, 1990) |
| | Explanation | yes (Chaudhury et al., 2021) | yes (Hussain and Cambria, 2018) | yes (Smolensky et al., 2016) |
| | Traceability | yes (Yabloko, 2020) | yes (Chen et al., 2020a) | yes (Smolensky, 1990) |
| | Summary | High | High | High |

