# OpenReview forum: "Evaluating Neuro-Symbolic AI Architectures: Design Principles, Qualitative Benchmark, Comparative Analysis and Results"
_nesyconf.org/NeSy/2025/Conference — NeSy 2025 Poster_

### Official Review · Reviewer_1MUA · 2025-03-25
**Review on an analysis of Neuro-Symbolic AI architectures**

**Rating:** 6
**Confidence:** 4

**Review:**

Summary
This paper analyzes various Neuro-Symbolic AI (NSAI) architectures across key aspects such as generalization, interpretability, and scalability. It compares different NSAI models, identifying the Neuro → Symbolic ← Neuro architecture as the most effective. A case study in 4D printing demonstrates the potential of NSAI in optimizing material design and manufacturing processes. The study underscores NSAI’s role in enhancing AI transparency and efficiency, with future research focusing on scaling and refining these architectures.

Strength
1. This paper provides a clear and thorough categorization of NSAI methods.
2. Based on scientific literatures, empirical findings, and design principles underlying each architecture, this paper conducts a comprehensive assessment of each class of methods.

Weakness
1. The assessment of the methods lacks detailed explanations and supporting literatures.
2. The “4D printing Application” remains in concept level, lacking convincing experimental results.

**Anonymity:**

Remain anonymous

---

### Official Review · Reviewer_KHhN · 2025-03-31
**Interesting setting, but unfounded results**

**Rating:** 2
**Confidence:** 5

**Review:**

The setting is very interesting. Indeed, the field of neurosymbolic AI is in desperate need of a good systematic comparison between systems.

However, this paper has some major problems.
Firstly, the authors give an overview of different neuro-symbolic architectures based on the taxonomy by Kautz. The authors mention that they expand on it, however, it is unclear from the text what the exact contribution of the authors is.
The major claimed contribution of this paper is an evaluation of these neurosymbolic categories along several (sub-)criteria. Although such an analysis would be nice to have, the shown results are completely unfounded. The authors do not mention which systems/papers were used in the analysis. They also give no indication of what the used evaluation criteria were for a system to be rewarded high, medium or low.
The authors also include a section on an application for 4D printing. This part seems tacked on, and the value of this to the remainder of the paper is questionable.

Overall, this leads me to strongly reject the paper in its current form. I think this paper would be far more valuable if the authors were to present the full results where the evaluation of each system and the rationale behind the scores are shown.

**Anonymity:**

Remain anonymous

---

### Official Review · Reviewer_YE1W · 2025-04-04
**Interesting Paper on" Evaluating Neuro-Symbolic AI Architectures: Design Principles, Qualitative Benchmark and Comparative Analysis" provides a nice overview over different architectures and principles" that would benefit from more depth and references.**

**Rating:** 6
**Confidence:** 3

**Review:**

The paper "Evaluating Neuro-Symbolic AI Architectures: Design Principles, Qualitative Benchmark and Comparative Analysis" provides a nice overview over different architectures and principles. However, both the overview and the evaluation stay superficial and would benefit from deeper insights in the architectures. The choice of discussed example applications seems random and would benefit from discussing different applications. The provided references to the archtitectures and even more to the evaluation are limited. For trusting the Table 1 evaluation I would expect more references and a deeper discussion and explanation for the table and the evaluations. The presented work is clear but at times as mentioned above superficial. Although the significance is limited the idea of an evaluation is valuable, if strengthened by more depth and related work. The choice of discussing the 4d printing application is not motivated thoroughly. The arrow notation to differentiate between different principles is hard to keep track of in this paper and therefore not easy to read in Table 1.
Pros:
- Clear (highlevel) writing.
- Nice idea to evaluate this.

Cons:
- Limited related work, references.
- limited and insufficiently motivated choice of discussed applications.
- Lack of in depth discussion and introduction to principles and evaluation.

My evaluation lacks a comparison with further submitted paper, since I only was assigned a single review.

**Anonymity:**

Disclose identity

---

### Official Review · Reviewer_PGho · 2025-04-11
**This paper summarizes the current Neuro-Symbolic AI Architectures quite well with a comparative analysis.**

**Rating:** 7
**Confidence:** 3

**Review:**

The authors have systematically summarized the methods of different Neuro-Symbolic AI architectures and highlighted some application areas where these architectures have been used. The paper is well-written and easy to read.

Their evaluation criteria are well-structured and cover several metrics, providing valuable insights into different methods. However, they do not cite or document any specific details regarding how they arrived at the evaluation results presented in Table 1. They only mention that they relied on scientific literature, empirical findings, and an analysis of the design principles underlying each architecture to conduct the evaluation. However, they do not specify which literature or findings they relied on for each individual architecture and evaluation metrics.

A more detailed description of the evaluation methodology would enhance the credibility of their work and be more helpful for researchers seeking to conduct further investigations.

**Anonymity:**

Remain anonymous